# Recent Advances in Herbal-Derived Products with Skin Anti-Aging Properties and Cosmetic Applications

**DOI:** 10.3390/molecules27217518

**Published:** 2022-11-03

**Authors:** Erika F. Costa, Wagner V. Magalhães, Luiz C. Di Stasi

**Affiliations:** 1Laboratory of Phytomedicines, Pharmacology, and Biotechnology (PhytoPharmaTech), Department of Biophysics and Pharmacology, Institute of Biosciences, São Paulo State University (UNESP), Botucatu 18618-689, SP, Brazil; 2Research and Development Department, Chemyunion Ltd., Sorocaba 18087-101, SP, Brazil

**Keywords:** skin anti-aging, natural products, cosmetic, herbal formulations

## Abstract

Although aesthetic benefits are a desirable effect of the treatment of skin aging, it is also important in controlling several skin diseases, mainly in aged people. The development of new dermocosmetics has rapidly increased due to consumers’ demand for non-invasive products with lower adverse effects than those currently available on the market. Natural compounds of plant origin and herbal-derived formulations have been popularized due to their various safe active products, which act through different mechanisms of action on several signaling pathways for skin aging. Based on this, the aim of the review was to identify the recent advances in herbal-derived product research, including herbal formulations and isolated compounds with skin anti-aging properties. The studies evaluated the biological effects of herbal-derived products in in vitro, ex vivo, and in vivo studies, highlighting the effects that were reported in clinical trials with available pharmacodynamics data that support their protective effects to treat, prevent, or control human skin aging. Thus, it was possible to identify that gallic and ferulic acids and herbal formulations containing *Thymus vulgaris*, *Panax ginseng*, *Triticum aestivum*, or *Andrographis paniculata* are the most promising natural products for the development of new dermocosmetics with skin anti-aging properties.

## 1. Introduction

Aging is a progressive decline in physiological function, leading to either age-related diseases or geriatric syndromes, including cardiovascular and obstructive pulmonary diseases, musculoskeletal disorders, several types of cancer, neurodegenerative diseases, and skin disorders [1,2]. Nowadays, the hypothesis that the aging process, age-related diseases, and geriatric syndromes share the same molecular and cellular mechanisms has been highlighted [2]. Together, all age-related diseases, physiological aging, and geriatric syndromes produce a heavy economic and psychological burden for patients and their families [1,3]. Among the various aging manifestations, skin aging is a process implicated with changes in skin appearance, aesthetic manifestations, and the development of several skin diseases. Skin aging is a natural, physiological, biochemical, and time-dependent process, resulting from a complex interaction between intrinsic and extrinsic factors, which produce cumulative deleterious changes in skin layers, morphology, physiology, and appearance [4]. Intrinsic factors are determined by genetic and physiological changes, whereas extrinsic factors are promoted by external stimuli, including ultraviolet (UV) radiation, diet, air pollution, abuse of tobacco and alcohol, lifestyle, toxins, and others [5]. Physiological skin changes affect the regulation of body temperature, fluid balance, loss of electrolytes and proteins, production of vitamin D, waste removal, sensory perception, immune response, and skin barrier function [5,6]. Aesthetic effects, such as wrinkles, skin spots, loss of elasticity, and thinning, affect the skin’s appearance and induce emotional, mental, and psychosocial problems [4]. On the other hand, the skin aging process is also closely related to an increased occurrence of cutaneous disorders and the development of dermatoses, benign and malignant tumors, itching, chronic wounds, dry skin, and skin depigmentation, which affect skin health and reduce the likelihood for a healthy aging process [4,6].

The skin has been recognized as the largest organ of the human body with a key role in the communication between the human body and the external world, including other organisms. The skin protects the body against mechanical and chemical damages, provides innate and adaptative responses, enables body thermoregulation, and acts as a sensory organ. Moreover, skin is important to one’s personal identity, closely related to one’s physical appearance, self-esteem, and self-consistency, which are basic self-concepts that define the consumption of cosmetics, personal care products, and other aesthetic products [7]. Self-esteem and self-consistency are the basis of consumers’ purchases of cosmetic products, with a global size of the cosmetics industry reported at USD 380.2 billion in 2019 and a projection that it will reach USD 463.5 billion by 2027 [8]. In the last few years, cosmetics and pharmaceutical companies have dramatically increased their development of new, safe, and efficacious drugs and therapeutical strategies to treat and prevent skin aging-related diseases [3]. Nowadays, several anti-aging strategies are available for dermatologists, but each one has advantages and disadvantages. Approaches to preventing and treating skin aging, including cosmetical skincare, correct sun protection, aesthetic non-invasive procedures, topical products (with antioxidants and cell regulator properties), invasive procedures (chemical peelings, radiofrequency, injectable skin bio-stimulators, and fillers among others), systemic agents (antioxidants and hormone therapy), and strategies to limit or reduce the exogenous factors of aging, such as the correction of one’s lifestyle and behavior habits, were recently revised [9]. On the other hand, consumers are concerned with their health and well-being, demanding non-invasive products with safety and efficacy based on natural bioactive products in cosmetics and other skincare products [5]. Although the bioprospection of active natural products, mainly of plant origin, has been overshadowed by the advent of new biotechnologies, the synthesis of new chemical compounds, and international regulatory systems for biodiversity access and conservation, plant biodiversity is a rich source of newly active molecules. The chemical diversity of compounds of plant origin includes different classes of secondary compounds, among which the following stand out: phenol, flavonoids and derivatives, anthocyanidins, anthocyanins, tannins, coumarins, terpenoids, stilbenes, and alkaloids compounds [10]. Moreover, plant species are also an extraordinary source of raw materials for the development of standardized herbal-derived products with scientific evaluations of their efficacy, safety, and quality control. They are useful in controlling many diseases, including the prevention and treatment of skin disorders, particularly skin aging processes.

In this review, we aimed to update and systematize the available data on the pharmacological activities of herbal-derived products evaluated by in vitro, ex vivo, and in vivo experimental studies, including clinical trials, providing data and insights for further studies and an evaluation of the potential applications of these active ingredients for the development of new cosmetic formulations. For this, an extensive review of the last 10 years was performed using the PubMed and Science Direct databases, targeting the main plant-based products potentially reported as anti-aging products, mainly those able to beneficially modulate the endogenous mediators and transcription factors associated with skin aging. Scientific evidence analysis was based on the use of in vitro, ex vivo, and in vivo studies, prioritizing those products with clinical trials and pharmacodynamic data. Although several selected products were based on plant extracts and different herbal formulations, chemically not characterized, those formulations here highlighted were fully evaluated in pharmacological studies and can be used for the development of herbal-derived products with skin anti-aging properties, mainly those reported as active products in clinical trials. Here, we summarize the main products of plant origin, highlighting those with efficacy, safety, quality control, and pharmacological actions on the main skin aging-related processes, such as skin elasticity, skin wrinkles production, skin hydration, skin pigmentation, and oxidative stress.

## 2. Skin Anti-Aging Natural Products by Acting on the Skin Elasticity and Wrinkle Formation

The skin is formed by the epidermis, dermis, and subcutaneous tissue, which are differentially affected by the aging process. However, the effects of skin aging are most evidenced on the dermis, which when atrophied by intrinsic and extrinsic factors promotes wrinkles and reduces skin elasticity. Dermal atrophy is closely related to reduced amounts of the extracellular matrix (ECM), including collagen, elastin, proteoglycans, glycosaminoglycans, fibronectin, and other glycoproteins [5].

Collagen, the major component of the ECM, can be divided into the fibrillar family, including collagen types I, II, III, and V, and non-fibrillar collagens IV, VI, VII, VIII, and XIV, which do not form classic fibrils [5,11,12]. In human skin, type I collagen comprises 80 to 90% of the total collagen, while type III comprises 8 to 12%, and type V comprises less than 5% [11]. In the dermis, the collagen produced by fibroblasts from procollagen (Figure 1) forms fibers, which maintain the strength and firmness of the skin [11,13]. In the aging process of the skin, a drastic loss of collagen fibers in the dermis occurs, leading to the formation of wrinkles and sagging, and the loss of skin viscoelasticity [13]. The degradation of collagen fibers is initiated by a proteolytic matrix metalloproteinase (MMP) enzyme, the MMP-1 [11]. After its cleavage, the collagen fiber can be further degraded by MMP-3 and MMP-9 [11]. Moreover, the loss of collagen fibers in aged skin is associated with an imbalance of collagen synthesis and degradation by MMP-1, indicating that the promotion of collagen synthesis in dermal fibroblasts and metalloproteinase inhibition are effective approaches to prevent and/or improve the symptoms of skin aging [13].

Elastin plays a crucial role in skin elasticity, returning the skin to its normal configuration after being stretched or deformed [5,11]. Elastin is broken down in the ECM by a serine proteinase called elastase, which increases with age and repeated UV radiation exposure (Figure 1). Since elastin has a key role in skin structural integrity and elasticity, its depletion contributes to wrinkle formation and skin aging, whereas the inhibition of elastase activity has been reported as potential protection against photoaging and structural damage to the ECM [11,12,13]. Although elastin is commonly known for its structural role, it may also be associated with wound repair and regeneration [12]. Within skin wounds, proteases such as MMPs act on elastin to liberate peptides which can form fibers, leading to an accelerated fibroblast proliferation, increased type I collagen, and increased tropoelastin [12].

Hyaluronic acid (HA), a high molecular glycosaminoglycan, can fill the free spaces of the ECM with water, providing the skin with a firmer and younger appearance [5]. Hyaluronic acid production occurs in dermal fibroblasts, and it has also been speculated that keratinocytes also promote its production (Figure 1), while its degradation occurs by an enzyme called hyaluronidase. Hyaluronic acid is a complex glycosaminoglycan with a chain that is negatively charged containing carboxyl and sulfate groups responsible for maintaining the water tissue content, with a key role in the hydration process of skin [11].

Collagen and elastin in the dermal layer of the skin are susceptible to the action of proteolytic metalloproteinases, which are secreted by keratinocytes and dermal fibroblasts under several stimuli, such as oxidative stress, UV radiation, and cytokines, among others [14]. Figure 1 illustrates the main ECM components, including its main proteins, collagen, elastin, and hyaluronic acid, and the enzymes responsible for their degradation.

To date, MMPs are categorized into five subgroups based on their substrates and include collagenases, gelatinases, stromelysin, matrilysins, and membrane-type metalloproteinases. Collagenases are MMPs able to degrade fibrillar collagen types I, II, III, and V without unwinding the triple-helical assembly of the substrate, and include mainly MMP-1, MMP-8, MMP-13, and MMP-18 [15]. Gelatinases, such as MMP-2 and MMP-9, digest several ECM components, such as collagen type I and IV, whereas stromelysins, including MMP-3, MMP-10, and MMP-11, have a domain arrangement similar to those of collagenases, but do not cleave fibrillar collagen type I [15]. Matrilysins, such as MMP-7 and MMP-26, degrade type IV collagen, but not type I, and membrane-type MMPs, including MMP-14, MMP-15, MMP-16, and MMP-24, have an additional C-terminal transmembrane domain with a short cytoplasmic tail and are responsible for degrading fibrillar collagen type I [14,15]. In addition to these five subgroups of MMPs, there are several MMPs, such as metalloelastase (MMP-12), RASI-1 (MMP-19), enamelysin (MMP-20), and epilysin (MMP-28). MMP-12, known as macrophage metalloelastase, is secreted by macrophages and fibroblasts in response to acute UV radiation, representing the most effective MMP that affects elastin. It is also responsible for the activation of other pro-MMPs, such as pro-MMP-1, 2, 3, and 9 [14]. During the aging process, there is an increased expression of MMPs with a reduction in the number of dermal fibroblasts, leading to decreased collagen production with the consequent formation of skin wrinkles [16]. It has been reported that in senescent fibroblasts, the expression of the tissue inhibitor of metalloproteinases (TIMPs) is decreased and this imbalance between the activation and inhibition of MMPs is a key component in the pathophysiology of both intrinsic and extrinsic aging [17]. TIMPs are skin enzymes from the protease family composed of TIMP-1, TIMP-2, TIMP-3, and TIMP-4 [11]. MMPs and TIMPs are often regulated in coordination to control excess MMP activity (Figure 1). Nevertheless, in aged skin, the elevation of MMP levels is not followed by a corresponding increase in the levels of TIMPs [11]. However, it has been reported that exposure to UV light decreases the levels of TIMPs, which contributes to the degradation of the ECM and subsequent skin aging [11,17].

ECM protein alterations have been closely related to wrinkle formation, mainly by reduced collagen content and increased levels of metalloproteinases [16]. Moreover, the weakening binding between the epidermis and dermis also contributes to wrinkle formation due to the reduction in collagen VII content and the loss of fibrillin-positive structures [5]. Deep wrinkle formation is also closely related to photoaging, which compromises collagen and elastin productions as well as the skin moisture barrier function [5], as evidenced by other studies demonstrating that ultraviolet radiation exposure, smoking, high sugar consumption, and sleep deprivation contribute to wrinkle formation and increased skin roughness [16,18].

Several products of plant origin have been reported as anti-aging products modulating one or more mediators related to skin elasticity and wrinkle formation. In the last few years, potential anti-aging isolated compounds from plants were reported to improve skin elasticity by different pharmacological actions (Table 1). Some xanthones from *Garcinia mangostana*, mainly α-mangostin, reduced UVB-induced skin wrinkles and inhibited epidermal thickening in hairless mice, in addition to other pharmacological effects [19]. Phenol compounds from *Dendrobium loddigesii*, including batatasin III, increased collagen synthesis [20], whereas several limonoids from *Carapa guainensis*, mainly genudin, acetoxygenudin, andirolide H, hydroxygenudin methyl angolensate, and carapanoside C, promoted collagen synthesis without cytotoxicity [21]. Triterpenoids from *Eriobotrya japonica*, mainly ursolic, pomolic, colosolic, and methyl colosolic acids, exhibited anti-aging activity by stimulating collagen and hyaluronic acid synthesis [22]. A plant hormone derived from the jasmonate-induced expression of proteoglycan in the human reconstituted epidermis [23] and a by-product polysaccharide obtained from red ginseng (*Panax ginseng*) inhibited UV-induced MMP-1 through activator protein-1 (AP-1) [24]. Finally, ferulic acid improved skin elasticity with significant bleaching, anti-redness, smoothing, and moisturized activity in a clinical study [25].

On the other hand, plant-based extracts or differential herbal formulations have been also exhaustively studied in several in vitro models, with different pharmacological actions on several mediators of skin elasticity, according to the following classification:
Elastase inhibitory activity was determined after the use of the plant extracts prepared with *Eugenia dysenterica* [26], *Gastrodia elata* [27], *Litchi sinensis* [28], *Magnolia officinalis* [29], *Malaxis acuminata* [30], *Manilkara zapota* [14], *Nephelium lappaceum* [28], *Phyllanthus emblica* [14], *Sclerocarya birrea* [31], *Sylibum marianum* [14], *Spatholobus suberectus* [32], *Tamarindus indica* [28], and a polyherbal formulation containing *Nyctanthes arbor-tristis* leaf, unripe and ripe *Aegle marmelos* fruit pulp, and the terminal meristem of *Musa paradisiaca* extracts [33];An increase in elastin synthesis or gene expression was observed after treatments with plant extracts of *Piper cambodianum* [34] and *Bidens pilosa* [35];An increase in pro-collagen expression or synthesis was observed after treatments with plant extracts prepared from *Alchemilla mollis* [36], *Azadirachta indica* [37], *Camellia sinensis* [38], *Citrus junus* [39], *Trapa japonica* [40], and a mixture of plant extracts of *Kochia scoparia* and *Rosa multiflora* [41];An increase in collagen synthesis or expression was observed after treatments with plant extracts of *Andrographis paniculata* [42], *Cassia fistula* [43], *Camelia sinensis* [44], *Passiflora tarminiana* [45], *Physalis peruviana* [46], *Piper cambodianum* [34], *Solanum tuberosum* [13], and *Bidens pilosa* [35];The modulation of hyaluronic acid synthesis was observed after treatments with plant extracts of *Cassia fistula* [43], *Penthorium chinense* [47], and *Salvia officinalis* [48];The downregulation of MMP-1 expression was described after treatments with plant extracts of *Alchemilla mollis* [36], *Allium sativum* [49], *Azadirachta indica* [37], *Camellia sinensis* [38], *Gastrodia elata* [27], *Kochia scoparia* [41], *Magnolia officinalis* [29], *Passiflora tarminiana* [45], *Penthorium chinense* [47], *Rosa multiflora* ([41], and *Syzygium aromaticum* [50];The downregulation of MMP-2 expression was described after treatments with plant extracts of *Cassia fistula* [43], *Magnolia officinalis* [29], and *Pourthiaea villosa* [51];A reduction in other metalloproteinases was observed after treatments with plant extracts of *Penthorium chinense* [47], *Piper cambodianum* [34], *Pousthiaea villosa* [51] and *Syzygium aromaticum* [50];The inhibition of the unspecific collagenase activity was observed after treatments with plant extracts of *Cassia fistula* [43], *Curcuma heyneana* [52], *Eugenia dysenterica* [26], *Hibiscus sabdariffa* [53], *Litchi chinens* [38], *Magnolia officinalis* [29], *Malaxis acuminate* [30], *Manilkara zapota* [14], *Nephelium lappaceum* [28], *Passiflora tarminiana* [45], *Phyllanthus emblica* [14], *Piper cambodianum* [34], *Sclerocarya birrea* [31], *Sylibum marianum* [14], and *Tamarindus indica* [28].

Moreover, two clinical trials were performed to study herbal product formulations on skin elasticity. A study investigated the anti-aging potential in human skin of a topical treatment containing a mixture of *Camellia sinensis* leaf, *Polygonum cuspidatum* root, *Ginkgo biloba* leaf, *Cynara scolymus* leaf, and *Selaginella tamariscina* and reported beneficial effects on facial skin as well as an improvement in ECM proteins’ production, although the data are not statically significant [54]. An herbal preparation containing *Vaccinum vitis-idaea* and *Phyllanthus emblica* was orally administered to volunteers in a double-blind placebo-controlled clinical trial, producing significant beneficial effects on skin elasticity [55]. These effects were related to improvements in skin thickness and stratum corneum water content, and a reduction in facial wrinkles in a dose-dependent manner [55]. Moreover, a phytocosmetic formulation containing *Thymus vulgaris*, after its topical application in volunteers, produced a reduction in perioral and crow’s feet wrinkles, accompanied by a reduction in nasolabial and smile lines, and an improvement in face oval remodeling [4].

## 3. Skin Anti-Aging Natural Products by Acting on the Skin Oxidative Stress

Reactive oxygen (ROS) and reactive nitrogen (RNS) species production are physiological metabolic events. However, when there is an imbalance between the generation of ROS/RNS and the antioxidant defense system, a process occurs known as oxidative stress, which leads to the damage of lipids, proteins, and DNA, the dysregulation of cell signaling pathways, and altered cytokine release, resulting in the development of several diseases, including skin aging [10,56]. The excessive production of ROS/RNS is a common event occurring under the action of intrinsic and extrinsic aging factors. In intrinsic aging, ROS generation occurs by oxidative metabolism, whereas in extrinsic aging, ROS formation is due to repeated exposure to environmental damage factors, mainly UV radiation [56]. It has been established that several UV wavelengths, such as UVB, UVA-1, and UVA-2 radiation, are partly responsible for photoaging in human skin, nevertheless, UVA-11 rays are the major contributors because they penetrate deeper into the skin, reaching the dermis [18].

The main reactive species in the body include superoxide anion (^•^O_2−_), hydrogen peroxide (H_2_O_2_), hydroxyl radical (^•^OH), singlet oxygen (^1^O_2_), lipid peroxides, and nitrogen oxides [10], which are continuously produced as a by-product of aerobic cellular respiration [17]. There is an imbricated and complex interaction between reactive species production and several signaling pathways (Figure 2).

The increase in ROS production induces the synthesis of metalloproteinases (MMPs), which degrades the proteins of the extracellular matrix (ECM) via mitogen-activated protein kinases’ (MAPK) pathway activation [11,56]. MAPKs are a family of proline-directed Ser/Thr kinases, including extracellular signal-regulated kinases (ERKs), p38, and c-Jun NH2-terminal kinase (JNK). High reactive species levels induce the JNK signaling pathway, which activates the activator protein 1 (AP-1), a transcriptional factor, leading to the over-expression of several metalloproteinases (MMPs), mainly MMP-1, MMP-3, MMP-9, and MMP-12, with a consequent degradation in collagen and elastin [11,17,56]. Similarly, oxidative stress is closely related to the nuclear factor-kappa B (NF-κB) signaling pathway, a transcription factor that regulates the gene expression of growth factors, chemokines, cytokines, and cell adhesion molecules in healthy or numerous abnormal conditions, including aging [11,56]. The NF-κB pathway can be activated by reactive species, leading to increased tumor necrosis factor α (TNF-α) levels and MMP-1 and MMP-3 expression in dermal fibroblasts [11]. Oxidative stress can also induce skin inflammation, activating NF-κB signal transduction, since it is responsible for the expression of several inflammatory cytokines in the skin. Oxidative stress is also related to the inhibition of the transforming growth factor β (TGF-β) signaling pathway in dermal fibroblasts by activating AP-1 (Figure 2), compromising the collagen and fibronectin synthesis. As demonstrated in Figure 2, TGF-β binds to a TGF-β type II receptor (TβRII), which recruits and phosphorylates a TGF-β type I receptor (TβRI), leading to the activation of the transcription factors SMAD2 (SMAD family member 2) and SMAD3 (SMAD family member 3), which, once activated, combine with SMAD4 (SMAD family member 4) to form SMAD complexes [11]. Activated SMAD complexes translocate into the nucleus and interact with SMAD-binding elements (SBE) in the promoter regions of TGF-β target genes, regulating procollagen type I synthesis in human skin [11].

In another oxidative stress pathway, nitric oxide synthase (NOS) modulates the reaction between superoxide radicals and nitric oxide radicals produced by the enzymatic oxidation of arginine to citrulline [10]. The reaction generates peroxynitrite, a reactive species able to induce the nitrosylation of proteins and oxidation of lipoproteins, and via its decomposition produces nitrite and hydroxyl radicals, while nitric oxide by autooxidation generates nitrous anhydride, a source of nitrite [10]. The peroxynitrite in the presence of dioxide carbon produces nitroperoxycarbonate, which by the homolysis process leads to the formation of carbon trioxide and nitrogen dioxide [10]. 

The control of oxidative stress for the body’s defense against free radical tissue damage and macromolecules’ oxidation is modulated by different endogenous antioxidant systems classified into (1) nonenzymatic antioxidants, including micronutrient components, such as zinc, copper, iron, manganese, selenium, and (2) enzymatic endogenous systems, including superoxide dismutase (SOD), catalase (CAT), and glutathione (GSH)-related enzymes [10,57]. The CAT enzyme is responsible for catalyzing the conversion of H_2_O_2_ to one molecular oxygen and two molecules of water, whereas the SOD enzyme has a crucial role in the defense against oxidative stress through the detoxification of superoxide radicals into hydrogen peroxide (H_2_O_2_). Hydrogen peroxide is subsequently converted to water and oxygen by glutathione peroxidase (GPX) and CAT, thereby preventing ROS formation. In association with SOD and CAT, GPX plays an important role in the reduction of H_2_O_2_ and lipid peroxides to produce water and lipid alcohol, also contributing to the modulation of oxidative stress [10]. Thus, glutathione S-transferase (GST), glutathione reductase (GR), and γ-glutamyl transferase (γ-GT) enzymes act by different mechanisms to reduce the availability of free radicals and control oxidative stress [10]. Figure 3 illustrates the pathway activation of the major regulator of the endogenous antioxidant system, the nuclear factor erythroid 2 (NEF2)-related factor 2 (Nrf2), which protects cells from several stress agents via the regulation of the transcriptional activation of antioxidants genes [50]. Under normal physiological conditions, Nrf2 is found in the cytoplasm associated with its inhibitor Kelch-like ECH-associated protein 1 (Keap-1), an actin-binding protein [57]. Upon oxidative stress conditions, Keap-1 is rapidly degraded through the ubiquitin-dependent proteasome pathway, which eventually causes Nrf2 accumulation and transcriptional activity (Figure 3). After the dissociation of its inhibitor, Nrf2 translocates into the nucleus, where it can bind to antioxidant response element (ARE) in the promoter region of Nrf2 target genes, inducing the transcription of the response genes to produce several endogenous antioxidant modulators (Figure 3).

Several products of plant origin were reported as anti-aging plant products by modulating one or more pathways of skin oxidative stress, mainly protecting human skin against the photoaging process. In the last years, the antioxidant activity of natural compounds from plants or herbal-based products has been extensively investigated. However, most of these studies are based only on the determination of the free radical scavenging activity. Although several in vitro assays have been used, including DPPH (2,2-diphenyl-1-picrylhydrazyl), ABTS (2,2′-azinobis-(3-ethylbenzothiazoline-6-sulfonic acid), FRAP (ferric reducing antioxidant power), CUPRAC (cupric reducing antioxidant capacity), and ORAC (oxygen radical absorbance capacity), the majority of the tests have the same principle: they use a synthetic colored radical or redox-active compound to evaluate the ability of specific products to scavenge the radical or to reduce the redox-active compound. The most used methods were DPPH (2,2-diphenyl-1-picrylhydrazyl) and ABTS (2,2′-azinobis-(3-ethylbenzothiazoline-6-sulfonic acid) radical scavenging assays.

A DPPH assay is based on the reduction of the DPPH radical by a product that can donate a hydrogen atom to the DPPH radical, which initially is observed to be purple. This results in the reduced form of DPPH (1,1-diphenyl-2-picryl hydrazine), observed to be yellow, indicating potential antioxidant activity that is spectroscopically determined. On the other hand, an ABTS assay is based on the generation of a blue/green ABTS^+^ that can be reduced by antioxidants. In addition to free radical scavenging activity, some researchers have evaluated the action of plants and herbal-derived products on antioxidant enzymes’ expression and activity, such as CAT, SOD, GPX. 

In the last few years, isolated compounds of plant origin modulating oxidative stress have included a few compounds. Xanthones from *Garcinia mangostana*, mainly α-mangostin, were able to increase SOD and CAT activities in UVB-irradiated hairless mice [19]; gallic acid, a compound that commonly occurs in plants, promoted in vitro DPPH scavenging activity and inhibited lipid peroxidation [58,59]; kinetin increased the CAT level and reduced malondialdehyde (MDA), a marker of lipid peroxidation in mice [60]; and several phenolics from *Dendrobium loddigesii* were able to scavenge free radicals in DPPH assay [20]. On the other hand, several plant extracts and herbal standardized products were described as having antioxidant properties and being potentially useful as skin anti-aging products (Table 2). The antioxidant activity of these plant extracts was preferably described using either the radical scavenging activity in a DPPH assay, the ABTS, and the FRAP scavenging activity, or the decrease in UVB-induced ROS generation (Table 2). Among the plant extracts that modulated oxidative stress, it is important to highlight those herbal preparations with a range of antioxidant properties by different methods. *Camelia sinensis* also upregulated SOD, CAT, and GPX in human dermal fibroblasts and increased the NRf2 transcriptional level and nuclear translocation [38], whereas *Pourthiaea villosa* and *Ulmus macrocarpa* increased SOD1 and SOD 2 protein levels and inhibited the cellular senescence induced by hydrogen peroxide [51,61].

## 4. Skin Anti-Aging Natural Products by Acting on the Skin Pigmentation

Melanin pigmentation plays a key role in skin protection from the natural harmful effects of UV radiation because it absorbs approximately 50–75% of the UV radiation [64]. Thus, abnormal skin pigmentation is a common signal in chronologically and environmentally induced skin aging. Although it acts as a skin protector, melanin with an excessive production leads to hyperpigmentation disorders, including melasma, freckles, and skin age spots, mainly on the human face [65]. Skin pigmentation is closely determined by the melanin synthesis in melanocytes, melanosome transfer to keratinocytes, and melanosome degradation. Tyrosinase is the major enzyme involved in the several steps of melanin synthesis generating l-DOPA (l-3,4-dihydroxyphenylalanine) from l-tyrosine and dopaquinone from l-DOPA (Figure 4), which, after another biochemical reaction, produces two types of melanin, the brown-black eumelanin and yellow-red pheomelanin [65]. Skin pigmentation is a complex process involving several factors, mainly oxidative stress, DNA damage, hormonal changes, and autophagy impairment, which are also related to the skin aging process and skin feature changes. Besides tyrosinase (TYR), the melanogenesis process involves the tyrosinase-related protein-1 (TYR-1) and tyrosinase-related protein-2 (TYR-2). Melanin synthesis initiates when TYR catalyzes the hydroxylation of l-tyrosine to l-DOPA and oxidation of l-DOPA to dopaquinone, which is the basic substrate for both eumelanin and pheomelanin synthesis [65]. When cysteine is available, a reaction with dopaquinone produces 3-or-5-cysteinyl DOPAs, which in turn, results in pheomelanin [65,66]. On the other hand, dopaquinone can form dopachrome, which by spontaneous decarboxylation generates 5, 6-dihydroxyindole (DHI), and after oxidation and polymerization, produces dark brown/black polymers, known as DHI-melanin. However, if TYRP-2 is available, dopachrome tautomerizes without losing its carboxylic acid, producing DHI-2-carboxylic acid (DHICA), which can oxidize and polymerize to form DHICA-melanin or a light brown color (Figure 4) [65].

Melanogenesis is regulated by several signaling pathways and transcription factors, including tyrosine kinase receptor, its ligand stem cell factor (SCF), and microphthalmia-associated transcription factor (MITF), which is the major regulator of melanogenesis that after activation induces TYR, TYRP-1, and TYRP-2 enzymes [47,66]. Some signaling pathways regulate MITF expression (Figure 5), including the melanocyte-specific melanocortin-1receptor (MCR1), predominantly expressed in melanocytes, which can be activated by melanocyte-stimulating hormone (α-MSH) and adrenocorticotropic hormone (ACTH), able to cleavage pro-opiomelanocortin [65,66]. When α-MSH binds to MCR1, it stimulates adenylyl cyclase, leading to an increase in the intracellular concentration of cyclic adenosine monophosphate (cAMP) with the subsequent activation of protein kinase A (PKA), which in turn, phosphorylates cAMP response element (CREB) protein, a transcription factor of several genes that, when associated to MITF, induce the production of TYR, TYRP-1, and TYRP-2 [65,66].

UV radiation is the main environmental trigger factor of skin photoaging and hyperpigmentation. UV exposure increases the expression of some key mediators of melanogenesis and activates several signaling pathways related to skin pigmentation, resulting in increased melanin production [53]. UV radiation activates p53 protein, which mediates the transcription of the POMC (pro-opiomelanocortin) precursor gene [65,66]. POMC gives rise to α-MSH and regulates pheomelanin and eumelanin production, activating signal transduction and inducing the MITF [65,66]. The enhanced MITF expression increases the expression of MITF-regulated proteins, such as TYR, TYRP-1, and TYRP-2, leading to increased melanin production and protease-activator receptor 2 (PAR-2) expression, stimulating the transfer of the melanosome from melanocytes to keratinocytes [65]. Thus, the autocrine secretion of ACTH, α-MSH, and endothelin-1 is also induced by UV exposure, through the expression of interleukin-1 (IL-1) in human keratinocytes [66]. ACTH and α-MSH can activate MCR1, increasing the expression of melanogenic enzymes and consequently the melanin synthesis. UV radiation exposure also induces prostaglandin 2 release by human melanocytes conducive to TYR modulation via subtypes of G protein-coupled receptors regulating the cAMP/PKA signaling pathway [66].

A few isolated products of plant origin reduce melanin production by acting as skin lightening agents through different mechanisms, including phenols from *Dendrobium loddigesii* and triterpenes from *Erobotrya japonica*, both acting by in vitro inhibitory action on the tyrosinase activity [20,22], whereas gallic acid acts by inhibiting l-DOPA oxidation [59]. On the other hand, plant-derived products acting on skin pigmentation include plant extracts from several medicinal and aromatic plants (Table 3). An inhibitory action of in vitro tyrosine activity was achieved by *Cassia fistula*, *Citrus jonus*, *Curcuma heyneana*, *Eugenia dysenterica*, *Hibiscus sabdariffa*, *Litchi sinensis*, *Malaxis acuminata*, and *Prosopis cineraria* in different tyrosine evaluation assays (Table 4). Plant extracts obtained from *Citrus jonus*, *Litchi sinensis*, *Magnolia officinalis*, *Penthorum sinensis*, *Prosopis cineraria*, and *Tamarindus indica* produced effects reducing the total content of melanin in different cell cultures (Table 4). In addition, *Hibiscus sabdariffa* downregulated the gene expression of several biomarkers of skin pigmentation, such as MITF, tyrosinase, TRP-1, and TRP-2 [53], whereas plant extracts containing *Litchi sinensis* and *Tamarindus indica* inhibited TRP-2 activity in melanoma cells and downregulated tyrosinase expression, respectively [28].

## 5. Skin Anti-Aging Natural Products by Acting on the Skin Hydration

The skin aging process is also induced by a decline in cell barrier function accompanied by increased epidermal fragility, thinning, and skin dryness or xerosis [67]. The barrier function is modulated by the stratum corneum and cell junctions that control transepidermal water loss (TEWL), which is directly affected by the lipid composition present in the stratum corneum, leading to water loss mainly from keratinocytes [59,67]. 

The maintenance of skin moisture is also partially modulated by natural moisture factor content in keratinocytes, which can improve the stratum corneum’s plasticity. The interplay between skin hydration and skin aging is an imbricated process related to high or constant exposure to environmental factors, including UV radiation. UV radiation exposure increases TEWL and downregulates involucrin, filaggrin, and loricrin, key proteins of the cornified cell envelope, which is an important constituent of the skin barrier [19,67]. Thus, it has been suggested that a product able to improve skin hydration, prevent skin dryness, and reduce TEWL is an important agent for the maintenance of skin cell functions [59]. 

Recently, several skin moisturizers have been proposed as complementary agents for skin aging improvement, mainly natural isolated compounds of plant origin and some herbal preparations. *Triticum aestivum* extract oil containing several polar lipids was evaluated in a randomized placebo-controlled clinical trial and produced an improvement in facial hydration, skin roughness, and skin radiance when compared with the placebo group [68]. In a clinical trial, ferulic acid, a phenol that commonly occurs in several plants, demonstrated important moisturizing activity, increasing skin hydration level and improving several anti-aging effects, such as anti-redness, smoothing, and skin elasticity [25]. Xanthones from *Garcinia mangostana* upregulated involucrin, filaggrin, and loricrin gene expression in the skin tissues of UVB-irradiated mice [19]. In a clinical trial, the extract of *Fragaria ananassa* enriched with ascorbic acid improved skin hydration and elasticity accompanied by a reduced skin tone [69]. In addition, *Ginkgo biloba* plant extract increased the TEWL after topical application in mice, whereas a mixture of *Vaccinum vitis-idaea* with *Phyllanthus emblica* improved the stratum corneum water content in a clinical trial study [55,70]. In another clinical trial, kinetin, a plant hormone responsible for regulating growth and differentiation, improved the skin’s moisture content in human skin after a few weeks of using a kinetin-containing cream, as well as the TEWL and skin texture [71]. A recent study demonstrated an *Anadenanthera colubrina* polysaccharide-standardized herbal preparation increased the expression of aquaporin 3, filaggrin, and involucrin with a simultaneous reduction in the TEWL in ex vivo human skin fragments and a clinical trial [72].

## 6. Skin Anti-Aging Natural Products by Acting on other Signaling and Transcriptional Pathways

The main signaling pathways related to the skin aging process induced by UV radiation and other environmental damage stimuli include MAPK, AP-1, Nrf2, NF-κB, SIRTs (sirtuins), TGF-β, and integrins. Some of these signaling pathways are imbricated with several factors affecting skin aging, mainly oxidative stress, skin pigmentation, skin hydration, and wrinkle formation. This way, products affecting these signaling and transcriptional pathways are potentially useful to control skin aging processes.

MAPKs are an important signaling pathway activated by UV radiation and play an important role in the regulation of several cell functions, such as cell proliferation, cell differentiation, apoptosis, and inflammation [32]. The MAPK family is composed of some proteins, including JNK, p38 kinase, and ERK, which constitute the serine/threonine kinase family. Exposure to UVB irradiation promotes the phosphorylation of ERK, p38, and JNK in hairless mice, whereas the expression of JNK and p38 was also increased in cell cultures after an H_2_O_2_ treatment and UVB radiation exposure [16,19,32]. The activation of ERK promotes proliferation and cellular growth, while the activation of JNK by ROS leads to p53 activation and premature senescence [16]. The MAPK pathway (Figure 6) is also responsible for regulating the AP-1, which in turn up-regulates MMPs’ expression, leading to the degradation of collagen (Figure 6). Moreover, the activation of Nrf2 and NF-κB is also mediated by the MAPK pathway (Figure 6). AP-1 and its constituent proteins, c-Jun and c-Fos, were activated by UV radiation by an increase in the protein phosphorylation in fibroblasts [36,37,50]. Moreover, AP-1 transactivation was increased in response to UV exposure, promoting the skin aging process, since this transcription factor up-regulates MMPs, leading to the degradation of collagen [24]. 

The MAPK pathway may trigger Nrf2 activation (Figure 6) [36,38,50]. UVB radiation also increased the Nrf2, NAD(P)H: quinone acceptor oxidoreductase 1 (NQO1) and heme-oxidase 1 (HO-1) production in the cell culture of fibroblasts, whereas agonists of Nrf2 improved the NQO1 and HO-1 gene expression [36,38]. Similarly, the MAPK signaling pathway also activated the transcription factor NF-κB. Moreover, the NF-κB pathway is also activated in exposure to exposure to UVB radiation, leading to the phosphorylation of p65, which is an active unit of NF-κB that increases the level of TNF-α, IL-1, and IL-6 and the expression of MMPs (Figure 6), key mediators of the skin aging process [38,73]. Besides MAPK, NF-κB, AP-1, and related signaling pathways, senescence-associated *beta*-*galactosidase* (*SA*-*β*-*gal*), SIRTs, TGF-β receptor, peroxisome proliferator-activated receptor γ (PPAR-γ), vascular endothelial growth factor (VEGF), and integrin β1 have been reported as pharmacological targets for the action of anti-aging products [17,42,49]. 

Several natural compounds and herbal preparations are potentially useful for the development of anti-aging products modulating different signaling pathways as demonstrated in Table 4, which were evaluated using in vitro assays with different cell types, including normal human dermal fibroblasts, human keratinocytes, 3T3-L1 embryonic fibroblasts, and human epidermal stem cells. Some of these plant extracts or isolated compounds were previously cited here as acting by other mechanisms of action to promote skin anti-aging effects.

**Table 4 molecules-27-07518-t004:** Main isolated compounds and plant-herbal products with anti-aging effects acting on UV-induced signaling pathway changes.

Plants/Products	Effects	Refs
*Alchemilla mollis*	In vitro (NHDF cells) inhibition of AP-1 activation, c-Jun, and c-Fos levels, and increase in Nrf2 pathway	[36]
*Allium sativum*	In vitro (HaCaT cells) inhibition of UV-induced increase in SA-β-gal levels and UV-induced decrease in SIRT1 activity	[49]
*Andrographis paniculata*	In vitro (Human epidermal stem cells) increase VEGF production and upregulation of integrin β1	[42]
*Azadirachta indica*	In vitro (NHDF cells) downregulation of c-Jun and c-Fos proteins and upregulation of TGF-β	[37]
*Camelia sinensis*	In vitro (HaCat cells) downregulation of HO-1 and upregulation of Nrf2 via phosphorylation of p38 and ERK	[38]
Catechin from *Leontopodium alpinum*	In vitro (HaCaT cells) suppression of p65 (NF-κB) phosphorylation	[73]
*Kochia scoparia*	In vitro (NHDF cells) upregulation of PPAR-γ and PPAR-α	[41]
Polysaccharides from *Panax ginseng*	In vitro (HaCaT cells) decrease UV-induced AP-1 transactivation	[24]
*Pourthiaea villosa*	In vitro (NHDF cells) inhibition of AP-1 activation-related JNK and p38 MAPKs	[51]
*Rosa multiflora*	In vitro (NHDF cells) upregulation of PPAR-γ and PPAR-α	[41]
*Spatholobus suberectus*	In vitro (HaCaT cells) downregulation of NF-κB and AP-1 pathways, inhibition of phosphorylated ERK1/2 and p38	[32]
*Solanum tuberosum*	In vitro (NHDF cells) induction of TGF-β signaling pathway, increment of phosphorylation of Akt and ERk1/2	[13]
*Syzygium aromaticum*	In vitro (NHDF cells) reduction of AP-1 signaling pathway via reduction in c-Jun and c-Fos levels, suppression of NF-κB expression, and upregulation of Nrf2	[50]
*Trapa japonica*	In vitro (NHDF and HaCaT cells) activation of the TGF-β/GSK-3β/β-catenin pathway	[40]
*Thymus vulgaris*	In vitro (3T3-L1 embryonic fibroblasts) upregulation of PPARγ and increase in adiponectin production	[4]
*Ulmus macrocarpa*	In vitro (HDF cells) blockade of JNK and p38 MAPK signaling	[61]
Verbascoside from *Syringa vulgaris*	In vitro (HaCaT cells) suppression of p65 (NF-κB) phosphorylation	[73]
*Vitis vinifera*	In vitro (NHDF cells) induction of SIRT 1 expression	[74]
Xanthones from *Garcinia mangostana*	*In vivo* (hairless mice) suppression of UVB-induced phosphorylation of MAPKs (ERK, P38, and JNK) and downregulation of IL-1β, IL-6, and TNF-α	[19]

AP-1 = Activator protein 1; ERK = Extracellular signal-regulated kinases; IκB-α = Inhibitor of nuclear factor kappa B; JNK = c-Jun NH2-terminal kinase; NF-κB = Factor nuclear kappa B; Nrf2 = Nuclear factor (erythroid-derived 2)-like 2; PPAR = Peroxisome proliferator activated receptor; SA-β-gal = Senescence-associated β-galactosidase; SIRT = Silent mating type information regulation 2 homolog; T*β*R = TGF-*β* receptor; TGF-β1 = Transforming growth factor-β1; VEGF = Vascular endothelial growth factor.

## 7. Promising Natural Compounds and Herbal Preparation for the Development of New Skin Anti-Aging Cosmetics

In order to select natural compounds of plant origin or herbal preparations, further scientific evidence is needed to support new studies for the development of innovative products with skin anti-aging properties. It is important to note that relevant compounds of plant origin, such as resveratrol, curcumin, quercetin, silymarin, and bakuchiol, as well as several herbal formulations containing rosehip (*Rosa moschata*) oil, bilberry (*Vaccinium myrtillus*) extract, licorice (*Glycyrrhiza glabra)* root extract, or pomegranate (*Punica granatum*) seed oil, were not discussed in this review, since these compounds or plant extracts have already been incorporated in a lot of market products claiming skin anti-aging properties. Based on this review of natural products of plant origin, it was possible to select some products while considering the quality of data available for each compound or herbal preparation described here. In the last 10 years, those compounds evaluated by clinical trials represent the best and most promising options for the development of new anti-aging products, including ferulic acid and gallic acid. Although multi-herbal preparations are promising formulations evaluated in clinical trials, we also selected some herbal preparations or standardized plant extract formulations based on one plant species as important sources for the development of skin anti-aging products, including *Thymus vulgaris, Panax ginseng*, *Triticum aestivum*, and *Andrographis paniculata*.

### 7.1. Ferulic Acid

Ferulic acid, also known as 4-hydroxy-3methoxycinnamic acid (Figure 7), is a very common phenol compound derived from hydroxycinnamic acid in several herbal foods, mainly fruits, vegetables, and beverages, such as coffee and beer, as well as in many medicinal plants. Ferulic acid has been proposed as a potent antioxidant compound due to its strong capacity to scavenge free radicals and activate cell responses against stress, with effects on cyclooxygenase 2, lipoxygenase, iNOS, SOF, CAT, and HO-1 activities as well as on several members of heat shock proteins, such as Hsp70 [75]. Ferulic acid has been evaluated and proposed as an important natural product useful in several disorders, including Alzheimer’s disease, Parkinson’s disease, cancer, cardiovascular diseases, diabetes, inflammatory processes, hepatic disorders, influenza, and skin disorders [75,76,77]. These protective effects are closely related to the antioxidant properties of ferulic acid on the target molecules related to oxidative stress as well as due to its effects on several signaling pathways, including the inhibition of nuclear β-catenin accumulation and Nrf2 activation, the induction of p38α MAPK, the mammalian target of rapamycin (mTOR) inhibition, and metalloproteinases inhibition, mainly MMP-9 [76,78]. 

On the skin, ferulic acid has been proposed as an important protective natural compound against skin disorders, mainly those related to UV exposure. Since UV-induced skin damage is closely related to ROS formation, the antioxidant properties of ferulic acid protect skin from environmental damage, mainly acting as a strong UV absorber and decreasing hydrogen peroxide via the up-regulation of HO-1 and Hsp70 in human dermal fibroblasts [77,78,79]. In human dermal fibroblasts, ferulic acid-induced keratin production via the inhibitory activity of β-catenin with a simultaneous improvement in wound-induced inflammation via the activation of the Nrf2 signaling pathway in mice [78]. A similar study using human dermal fibroblasts stimulated by UVA radiation demonstrated that ferulic acid significantly increased cell proliferation and cell cycle progression, and had effects related to antioxidant properties via the modulation of SOD and CAT gene expression [80]. In addition, when ferulic acid was assessed in mice with atopic dermatitis induced by 2,4-dinitrochlorobenzene (DNCB), it promoted potent anti-inflammatory activity, inhibiting Th2 cytokines and IgE and downregulating TNF-α, IL-4, IL-6, and IL-12 expression, effects that alleviated the DNCB-induced atopic dermatitis [81]. These photoprotective effects were also corroborated when ferulic acid was associated with two UV filters, ethylhexyl triazole and vis-ethylexyloxyphenol methoxyphenyl triazide [82]. In this study, a synergistic effect between ferulic acid and UV filters significantly increases both the sun protection factor and the UVA protection factor [82]. B16F10 melanoma cells and CCD-986sk fibroblasts were targeting ferulic acid to demonstrate a whitening effect via the inhibition of melanin synthesis, tyrosinase expression, and MITF expression as well as anti-wrinkle activity, inducing procollagen, hyaluronic acid, and TIMP synthesis, parallel to the inhibition of MMP-1 and MMP-9 expressions [83]. In chemically induced skin carcinogenesis in mice, oral ferulic acid completely prevented tumor formation but did not produce effects after topical application, and these suppressive effects on cell proliferation were related to the ability of ferulic acid to modulate lipid peroxidation and detoxication agents during the chemical carcinogenesis process [84]. Ferulic acid is promptly absorbed by the skin when topically applied and protects the skin from UVB radiation-induced skin erythema and pigmentation in cells from human breast reduction surgeries [85]. Docking studies indicated that ferulic acid and its derivative, 5-hydroxyferulic acid, interact with the active site of the lipoxygenase (LOX) enzyme, with a consequent inhibition of its activity, representing a potential mechanism of skin protection against UV radiation-induced oxidative stress [86]. Recently, ferulic acid was also evaluated in an elegant split-face comparative study, alone or associated with microneedling [25]. Ferlic acid induced a significant improvement in the skin’s elasticity, and promoted bleaching, anti-redness, smoothing, and moisturizing activities with better effects when associated with microneedling [25]. However, these protective effects were associated with mild adverse effects, including skin pruritus, a burning sensation, scaling, and erythema [25]. 

Since ferulic acid has good topical absorption by the skin at acidic and neutral pHs, this phenol compound can be absorbed by the skin in either dissociated or non-dissociated forms [85], with a high bioavailability and ability to exert its pharmacological effects. In addition, ferulic acid has been used as a stabilizer or adjuvant of several topical formulations containing vitamin C, vitamin E, azelaic acid, mandelic acid, phytic acid, or resorcinol with better skin protective effects when compared with the use of ferulic acid alone [75,87,88]. Moreover, ferulic acid can upregulate HO-1 and Hsp70 expression, inducing hormesis, which is represented by the mild stress-induced stimulation of protective mechanisms in cells and organisms resulting in beneficial effects on aging, mainly by maintaining homeostasis in oxidative stress disorders and youth cell morphology as well as by improving lifespans [77,89].

Although ferulic acid is already used in some cosmetic formulations available on the market as a skin anti-aging product due to its skin photoprotection, skin lightening, and anti-acne effects [90], the recent pharmacological data described above provide new scientific evidence supporting this compound as a natural active ingredient for the development of new dermocosmetical products to prevent skin aging, as well as to reduce or treat hyperpigmentation and atopic dermatitis. Ferulic acid’s pharmacokinetic and pharmacodynamic aspects, mainly its oxidative stress conditions as well as its health applications and toxicologic effects, have been extensively reviewed [75,76,77,88,90].

### 7.2. Gallic Acid

Gallic acid, the 3,4,5-trihydroxybenzoic acid (Figure 7) is also a phenol compound abundantly available in the bark, wood, leaves, fruit, seeds, and roots of vegetables, nuts, fruits, and medicinal and aromatic plants. It is found mainly in strawberries (*Fragaria ananassa*), grapes (*Vitis* species), bananas (*Musa* species), blueberries (*Vaccinium* species), apples (*Malus domestica*), walnuts (*Juglans regia*), cashews (*Anacardium occidentalis*), hazelnuts (*Corylus* species), black tea (*Camellia sinensis*), and Brazilian pepper trees (*Schinus terebinthifolius*). Gallic acid displays antioxidant and free radical scavenging properties, particularly anti-diabetic, anti-inflammatory, antiviral, and anti-tumor activities when used in different formulations, including in nanoparticles [58,59,91,92]. 

Several studies have reported the potential use of gallic acid in topical and oral formulations as skin anti-aging products. A clinical trial of a gel containing gallic acid and other phenol compounds from *Terminalia chebula* produced skin anti-aging activity in human volunteers, as evidenced by several changes in skin elasticity and a decrease in skin roughness [59]. Since the cosmetic application of gallic acid has a low solubility and bioavailability [58], studies have been performed with different formulations containing gallic acid, mainly those related to the production of gallic acid glycosides or nanoparticle formulations. Recently, it was reported that gallic acid glycoside produced by transglycosylation demonstrated a potent antioxidant and anti-tyrosinase product useful as a cosmetic ingredient with skin-whitening and anti-aging properties [58]. A gallic acid glycosidic formulation inhibited the pollution-mediated MMP-1 protein, cytochrome P450 1A1 gene expression, and IL-6 protein secretion [93]. When associated with other compounds in a multifunctional skincare formulation, it produced protective effects on the skin, including skin brightening, anti-aging, and anti-inflammatory properties via the modulation of several skin disease markers [93]. Gallic acid loaded in cationic noisome has the highest skin anti-aging activity when compared with free gallic acid, promoting antioxidant properties, melanin suppression, the inhibition of tyrosinase, and tyrosinase-related protein-2 activities, and inhibitory effects on MMP-2 production in B16F10 melanoma cells [91]. Gallic acid has been incorporated into cosmetic textiles, and when it was encapsulated in poly-ε-capralactone microspheres, it exhibited antioxidant properties, inhibiting lipid peroxidation after topical applications on the skin of human volunteers [92]. Moreover, the gallic acid-loaded cosmetic gel formulation with good skin adhesiveness exhibited a strong antioxidant effect and an inhibitory action of lipid peroxidation in the stratum corneum, whereas in human dermal fibroblasts, gallic acid-coated gold inhibited MMP-1 expression through several signaling pathways, such as MAPKs, NF-κB, AP-1, and apoptosis signal-regulating kinase 1 (ASK1), a MAPK3 that activates the MAPK4 pathway [94,95].

Recently, in a mouse model of dermatitis-like skin inflammation, gallic acid improved the thickness and pathology of the ears of mice [96]. These protective effects were related to gallic acid’s ability to both downregulate TNF-α, IL-4, INF-γ, and IL-17 and to upregulate IL-10 and TGF-β expressions, suggesting gallic acid is a compound that, besides acting as an antioxidant compound, also modulates the Th17 immune response [96]. Free gallic acid on psoriasis-like skin disease in vivo and in vitro models reduced the mRNA expression of keratin 16 and keratin 17, two biological markers of psoriasis [97]. These protective effects of gallic acid on psoriasis were related to an unexpected downregulation of Nrf2, which targeted keratin 16 and keratin 17, ameliorating the psoriasis area, the severity index scores, and the epidermal hyperplasia induced by the psoriasis-like disease in mice [97]. Gallic acid was also reported as a protective agent for the treatment of keloids, a fibroproliferative disorder of the skin [98]. Gallic acid inhibited keloid fibroblast proliferation, migration, and invasion as well as downregulated MMP-1 and MMP-3 and upregulated the tissue inhibitors of MMP-1, effects associated with the suppression of the protein kinase B signaling pathway [98]. The antiproliferative effects are important to explain the anti-tumor activities described for gallic acid in cell cultures of human breast cancer, human leukemia cells, and melanoma [99,100,101].

### 7.3. Thymus vulgaris

*Thymus vulgaris* L., belonging to the mint family Lamiaceae and popularly known as thyme, was recently evaluated in a double-blind placebo-controlled clinical trial to control the skin aging process [4]. In this innovative study, a standardized phytocosmetic preparation containing *Thymus vulgaris* significantly reduced facial wrinkles and expression lines, promoting face oval remodeling in women volunteers after topical application [4]. These effects were closely related to the increased production of adiponectin and the upregulation of PPAR-γ expression, which produced a soft tissue augmentation similar to those induced by soft tissue fillers but using a phytocosmetic formulation by a topical route with no manifestations of skin reactions or discomfort [4]. This mechanism of action was related to the presence of carvacrol and thymol, positions isomers, which were reported as compounds able to improve PPAR- γ expression or act as PPAR- γ agonists [102,103]. 

Thymol, 2-isopropyl-5-methylphenol (Figure 7), is abundantly found in several plants, including *Thymus vulgaris,* other *Thymus* species, *Ocimum* mint, and *Origanum* plant species, among others [102]. This phenol compound possesses multiple functions and pharmacological activities with a potential use in preventing or controlling several disorders. Thymol exhibits antioxidant properties, acting as a free radical scavenger and increasing the enzymatic activity of several endogenous antioxidants, such as SOD, CAT, GPX, and GST [102]. Moreover, thymol has been described as anti-inflammatory, antimicrobial, and useful against several tumors, metabolic disorders, atherosclerosis, and cardiovascular, renal, lung, liver, autoimmune, gastrointestinal, and neurodegenerative diseases [102]. Pharmacological activities have been related to the ability of thymol to modulate several molecular targets, including PPAR-γ, phosphoAMP-activated protein kinase (pAMPk), PPAR-α, and PKA [102]. The topical application of thymol attenuated atopic dermatitis induced by *Staphylococcus aureus* membrane vesicles, decreasing the epidermal and dermal thickness and the infiltration of neutrophils, downregulating pro-inflammatory cytokine and chemokines, and reducing the de serum levels of immunoglobulin G and E [104]. The skin’s anti-inflammatory and anti-psoriatic activities were reported after the topical application of the lipid nanoparticles containing thymol to improve the drug delivery and stability and decrease toxicity [105]. This gel’s nanoparticle formulation produced in vivo anti-inflammatory and antioxidant properties in both croton oil and anthralin-induced inflammation, reducing edema and thickness as well as improving wound healing in the imiquimol-induced psoriasis-like inflammation in mice [105]. A similar study evaluated the properties of surface-functionalized poly (lactico-glycolic acid) nanoparticles loaded with thymol and demonstrated in vitro anti-inflammatory effects in HaCat cells, as well as antioxidant, antimicrobial, and wound-healing effects against acne, reducing TNF-α, IL-1β, IL-1α, IL-6, and IL-8 expression [106,107].

The PPAR-γ, a molecular target associated with skin aging, is also modulated by carvacrol [4,103,108]. Carvacrol, 5-isopropyl-2-metylphenol (Figure 7), has been related to similar sources as and the pharmacological activities of thymol [109,110]. Carvacrol also inhibited tyrosinase activity at different concentrations, increased skin thickness in wild mice dependent on the transient receptor potential vanilloid 3 (TRPV3), and promoted the proliferation of the human keratinocytes’ HaCaT cells [111,112]. These effects of carvacrol were closely related to the stimulation of TGFα release and an increase in the phosphorylation levels of EGFR, PI3K, and NF-κB [112].

However, it has been shown that the combined treatment of thymol and carvacrol at different rates has produced better effects than treatments with thymol or carvacrol alone [102,109], suggesting that the use of herbal formulations containing plants, such as *Thymus vulgaris,* is a good strategy for the production and the development of new cosmetic products with skin anti-aging activity.

### 7.4. Panax ginseng

*Panax ginseng* C.A. Meyer, belonging to the family Araliaceae and popularly known as ginseng, is a medicinal plant that has been used as an herbal product in the traditional medicine of several countries of Asia, mainly Korea, China, and Japan, to treat a lot of diseases for more than 2,000 years and has a series of pharmacological activities related to the presence of ginsenosides [113]. Ginseng is a rich source of chemical compounds, such as ginsenosides, polysaccharides, peptides, fatty acids, and phytosterols, which display several effects on lipid and carbohydrate metabolism and angiogenesis, as well as neuroendocrine, cardiovascular, immune, and neurodegenerative disorders [113,114]. On the skin, ginseng, ginsenosides, their derivatives, and other chemical compounds have been described to treat several skin disorders, inhibiting in vivo and in vitro melanogenesis, protecting the skin from UV radiation, promoting skin wound healing, anti-wrinkle formation, and skin hydration, along with an anti-atopic dermatitis effect, all of which are potentially useful for the development of new skin anti-aging products.

In addition to the downregulation of MMP-1 expression induced by ultraviolet light via AP-1 transactivation and its effects on skin elasticity and wrinkle formation [24], a photoprotective effect was also reported in a clinical trial using an enzyme-modified *Panax ginseng* extract prepared after the incubation of *Panax ginseng* ethanol extract with a ginsenoside-β-glucosidase [115]. After topical applications in 22 female volunteers, twice daily for 12 weeks, the formulation promoted several skin benefits, such as inhibitory effects against UVB-induced skin aging, evidenced by a reduction in the global photodamage score, total roughness, smoothness depth, and roughness average, with no adverse reactions during treatments [115]. These protective properties were attributed to the presence of ginsenosides Rg1 and F2 (Figure 7). Skin photoprotection has been also reported to ginsenoside Rb1 and compound K, a metabolite of ginsenoside Rb1 (Figure 7) [116,117], indicating that *Panax ginseng* is a rich source of several active compounds with skin anti-aging properties. These photoprotective compounds act by different mechanisms of action, including the inhibition of UV-induced DNA damage via the induction or upregulation of the nucleotide excision repair complex and the reduction of the expression, secretion, and activity of the MMP-1, MMP-2, MMP-9, and MAPKs signaling pathway [115,116,118]. Recently, the ginsenoside Rg3 (Figure 7) was reported to attenuate skin disorders by a new mechanism of action, downregulating the murine double minute 2 and hypoxia-inducible factor 1α signaling pathways [119], whereas a ginseng plant extract induced photoprotection in NIH-3T3 cells via the activation of the protein kinase B signaling pathway [120].

On skin pigmentation, ginseng in different formulations can inhibit melanogenesis by acting on the several components of melanin production, inducing a skin whitening effect by different mechanisms of action, mainly as an inhibitor of tyrosinase, α-melanocyte-stimulating hormone, granulocyte-macrophage colony-stimulating factor, MITF, and NF-κB, as well as protein kinase A, protein kinase C, and MAPKs signaling pathways [114,121,122]. The effects on the melanogenesis’ process were attributed to the presence in *Panax ginseng* of the ginsenoside Rb1, ginsenoside F1, ginsenoside Rh23, ginsenoside C-Y, ginsenoside Rf, and p-coumaric, salicylic and vanillic acids [114,123,124,125,126,127]. *Panax ginseng* extracts and active components have been also reported as useful products to treat or control atopic dermatitis, hair loss, acne, skin wound healing, and skin hydration [116,128,129,130]. Extensive reviews containing the pharmacological and chemical characterization of herbal formulations and isolated compounds from *Panax ginseng* as well as its main mechanisms of action related to its anti-skin aging properties (anti-wrinkle formation, increase in skin elasticity, inhibition of melanogenesis, skin hydration, anti-acne, anti-atopic dermatitis, and anti-hair loss) support this medicinal plant as a potential raw material and source of chemical compounds useful for the development of new skin anti-aging products.

### 7.5. Triticum aestivum L.

*Triticum aestivum* L., synonymous with *Triticum vulgare L.,* belongs to the Poaceae family and is known as wheat grass or common wheat. It is one of the oldest cultivated crops and is a raw material to produce malt and beer [131]. Traditionally used to treat several diseases and disorders, including oxidative stress, cardiovascular diseases, diabetes, and inflammatory processes, wheatgrass is also used as a drink, food, and dietary supplement [132,133]. Recently, a polar lipid extract oil obtained from the endosperm of *Triticum aestivum* was evaluated orally in a double-blind placebo-controlled trial with 64 healthy women volunteers for 12 weeks as well as in ex vivo human skin explants [68]. In the clinical trial, the wheat grain oil treatment promoted an improvement in the women’s crow’s feet wrinkles and skin hydration, whereas, in the ex vivo experiments, *Triticum aestivum* oil promoted a significant increase in collagen in UV-irradiated skin explants [68], suggesting the use of wheat grain oil as an skin anti-aging product. Besides its anti-wrinkle effects, *Triticum aestivum* was also evaluated for in vivo DNCB-induced atopic dermatitis in mice and in vitro human keratinocytes, attenuating atopic dermatitis-like symptoms [134]. The protective effects were related to the reduced levels of IgE and downregulated C-C motif chemokine ligand 5 (CCL5), macrophage-derived chemokine (MDC), and interferon-γ-induced protein 10 (IP-10) after oral administration in mice [135]. In HaCat cells, *Triticum aestivum* at different concentrations downregulated CCL5, MDC, and IP-10, inhibited the signal transducer and activator of transcription 1 (STAT1) phosphorylation induced by TNF-α and INF-γ, and increased the suppression of cytokine signaling 1 (SOCS1) gene expression, which exhibits inhibitory activity against STAT1 [134]. Moreover, wheat extract oil attenuated UVB-induced photoaging in human keratinocytes and hairless mice, acting via an increase in collagen synthesis and a decrease in the TEWL, and prevented the reductions of procollagen type I HA and ceramide, improving the skin barrier function [135]. In HaCat cells, the treatment with wheat oil also increased the hyaluronic acid and collagen synthesis, acting via the suppression of MMP-1, with clear skin benefits [135]. Finally, common wheat extract or nanovesicles isolated from wheat seeds exhibited skin wound-healing processes, acting on the cell proliferation and promoting wound repair [136,137,138]. The mechanisms of wound repair induced by *Triticum aestivum* were correlated to the modulation of new fibronectin synthesis and the upregulation of HA synthase and collagen type I in human dermal fibroblasts [136,137], as well as via the upregulation of 153 proteins and the downregulation of 72 proteins in the secretome, which coordinates the processes of cell–cell interaction, cell proliferation, differentiation, adhesion, and migration [139].

### 7.6. Andrographis paniculata

*Andrographis paniculata* (Burm.f.) Ness. is an herbaceous plant from the Acanthaceae botanical family with several ethnopharmacological uses and a wide distribution in India, China, and other countries of Southeast Asia [140]. *Andrographis paniculata* is a plant with high chemical diversity, containing diterpenoid lactones, such as the following: several andrographolides; flavonoids, mainly flavanones, flavones, and flavonols; terpenoids, phenolic acids, including ferulic, caffeic, vanillic, and chlorogenic acids; chalcones, xanthones, and volatile compounds [140,141]. Based on its rich chemical diversity, *Andrographis paniculata* has anti-inflammatory, analgesic, antipyretic, anti-oedema, antioxidant, antihyperglycemic, hepatoprotective, antibacterial, antiparasitic, antiviral, antitumoral, immunomodulatory, antihyperlipidemic, contraceptive, neuroprotective, and protective cardiovascular effects [140,141,142]. The anti-aging properties of *Andrographis paniculata* hydroethanolic extract 50% were reported using in vivo, ex vivo, and in vivo clinical studies with thirty-two healthy female volunteers [42]. *Andrographis paniculata* extract, after topical treatments over four and eight weeks in volunteers, significantly decreased skin wrinkles and skin sagging as well as increased the dermal density of facial skin [42]. These effects were correlated with the ability of *Andrographis paniculata* extract to renew the rate of epidermal cells, increasing the proliferation in the epidermal stem cells, an effect related to the upregulation of the integrin β1, which plays a key role in epidermal stem cell maintenance and proliferation [42]. Moreover, *Andrographis paniculata* extract increased the levels of VEGF [42]. Recently, *Andrographis paniculata* extract was reported as a photoprotective product against UV-A and UV-B radiation by sun protection factor determination related to the presence of flavonoid derivatives [143].

## 8. Conclusions

In the last 10 years, an extensive number of plant species, including foods and medicinal and aromatic plants, were pharmacologically evaluated in in vitro, ex vivo, and in vivo biological assays, including clinical assays focusing on the determination of the potential use of these plants as skin anti-aging products. These studies aimed to determine their pharmacological effects on skin wrinkle formation, skin elasticity, oxidative stress, skin pigmentation, photoaging, and skin hydration, as well as on the several transcriptional signaling pathways related to the skin aging process. Several studies and reviews have reported the main mechanisms of action and the signaling pathways modulated by different herbal formulations and isolated compounds of plant origin, which have corroborated previous experimental data. Based on this, this review identified and selected some compounds (ferulic, acid, gallic acid, thymol, carvacrol, ginsenosides, and others) and herbal formulations (*Thymus vulgaris*, *Panax ginseng*, *Triticum aestivum*, and *Andrographis paniculata*) as the most promising products and raw materials for use in the development of new dermocosmetics with skin anti-aging properties.

## Figures and Tables

**Figure 1 molecules-27-07518-f001:**
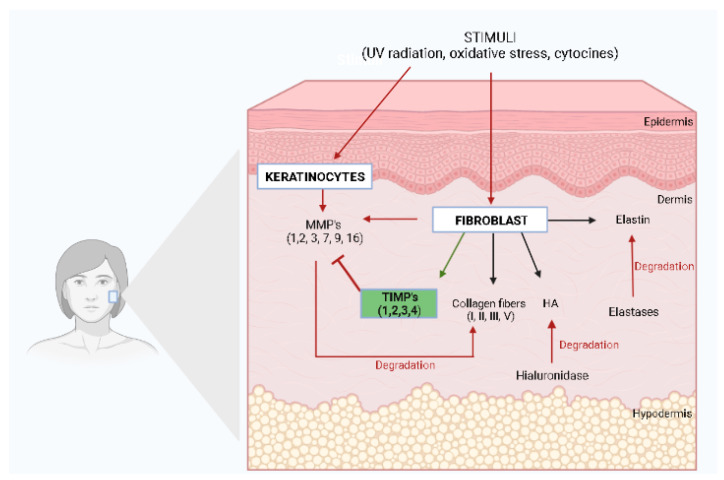
**Extracellular matrix components (ECM) and skin elasticity**. HA, Hyaluronic Acid; MMPs, Metalloproteinases; TIMPs, Tissue Inhibitor of Metalloproteinases. The illustration was drawn using BioRender software.

**Figure 2 molecules-27-07518-f002:**
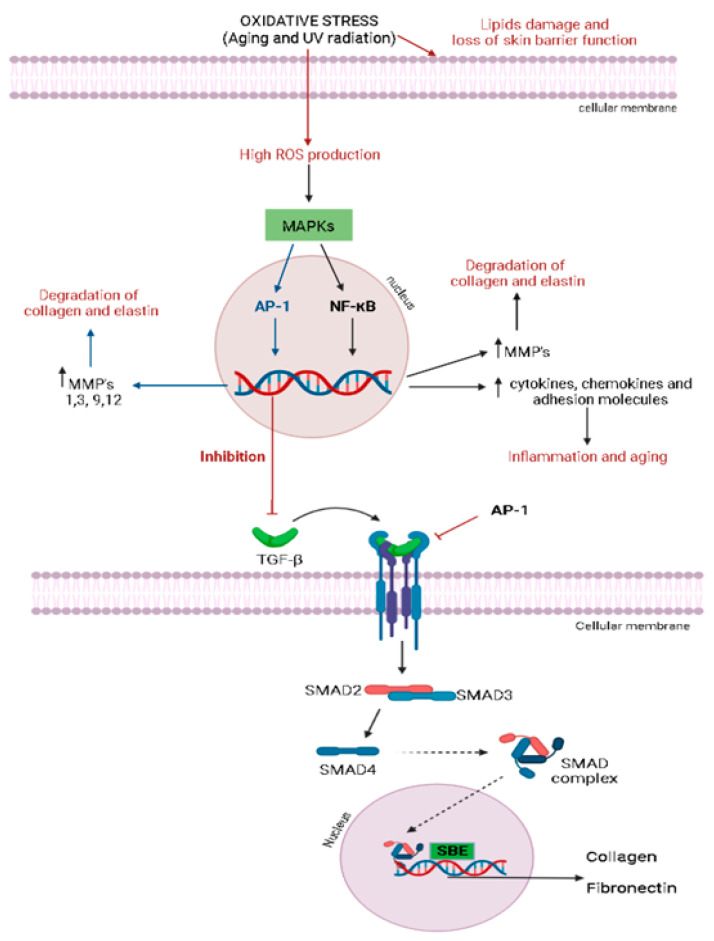
**TGF-β signaling pathway.** AP-1, factor activating protein 1; SBE, SMAD-binding elements; TGF-β, transforming growth factor beta; TβRI, TGF-β type I receptor; TβRII, TGF-β type II receptor. The illustration was drawn using BioRender software.

**Figure 3 molecules-27-07518-f003:**
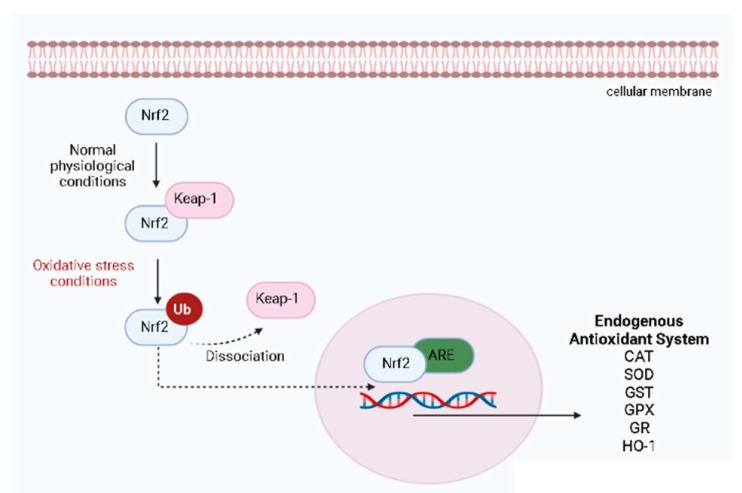
**Activation pathway of Nrf2 and the production of endogenous antioxidant molecules**. ARE, antioxidant response element; GST, glutathione S-transferase; HO-1, heme oxygenase-1; NQO1, NAD(P)H: quinone acceptor oxidoreductase 1; Nrf2, nuclear erythroid 2-related factor. The illustration was drawn using BioRender software.

**Figure 4 molecules-27-07518-f004:**
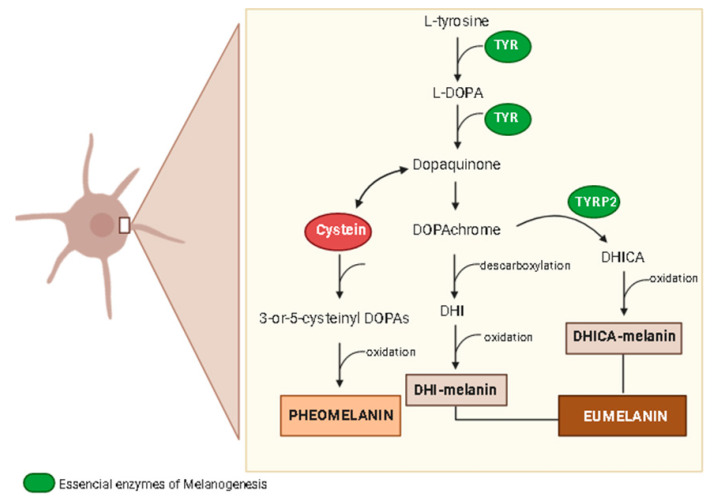
**Synthesis of melanin.** DHI, 5,6-dihydroxyindole; DHICA, DHI-2-carboxylic acid; l-DOPA, to l-3,4-dihydroxyphenylalanine; TYR, tyrosinase; TYRP2, tyrosinase-related protein-2. The illustration was drawn using Biorender software.

**Figure 5 molecules-27-07518-f005:**
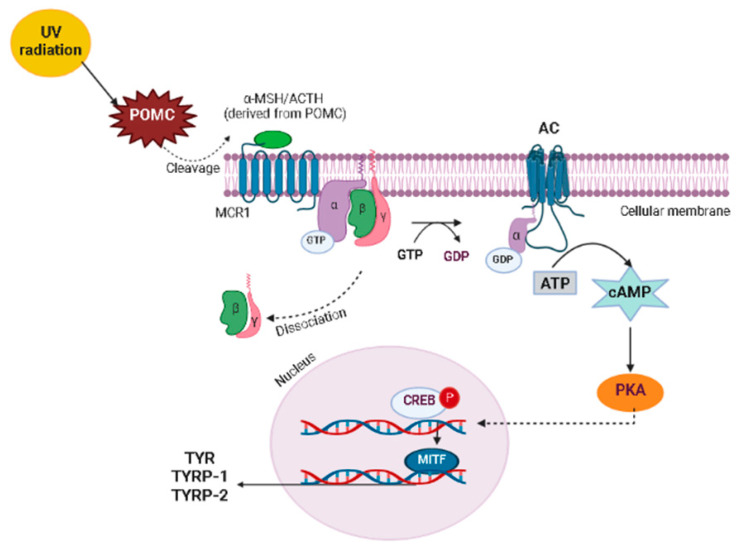
**Signaling pathway MITF expression.** α-MSH, melanocyte-stimulating hormone; AC, adenylyl cyclase; ACTH, adrenocorticotropic hormone; ATP, adenosine triphosphate; cAMP, cyclic adenosine monophosphate; CREB, cAMP response element; GDP, guanosine diphosphate; GTP, guanosine triphosphate; MCR1, melanocyte-specific melanocortin-1receptor; MITF, microphthalmia-associated transcription factor; PKA, protein kinase A; POMC, pro-opiomelanocortin; TYR, tyrosinase; TYRP-1, tyrosinase-related protein-1; TYRP-2, tyrosinase-related protein-2. The illustration was drawn using Biorender software.

**Figure 6 molecules-27-07518-f006:**
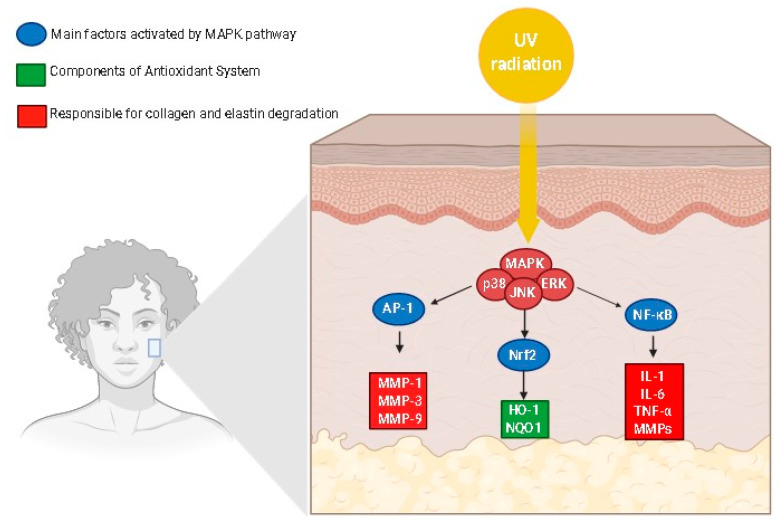
**Mitogen-activated protein kinase signaling pathway modulating mediators of the skin aging process.** AP-1, activator protein 1; Nrf2, nuclear factor (erythroid-derived 2)-like 2; NF-κB, Nuclear factor-kappa B; MMP, metalloproteinases; NQO11, NADPH-quinone oxidoreductase 1; HO-1, heme oxygenase-1. The illustration was drawn using Biorender software.

**Figure 7 molecules-27-07518-f007:**
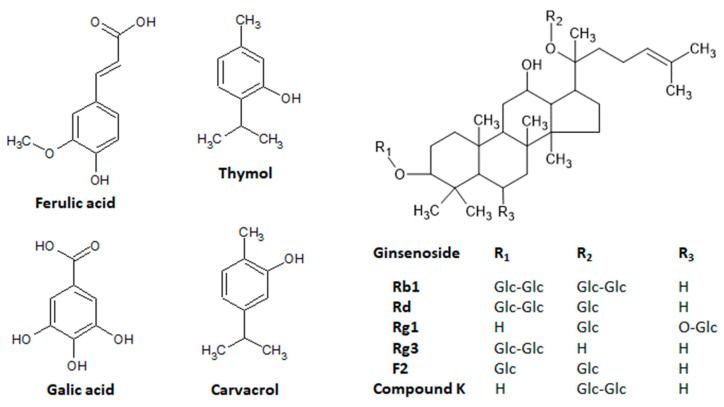
Main natural compounds of plant origin with promising skin anti-aging properties. Chemical structures were drawn using ACD/ChemSketch software.

**Table 1 molecules-27-07518-t001:** Main isolated natural compounds with anti-aging effects by acting on skin elasticity and wrinkle formation.

Compounds	Test	Effects	Refs
α-mangostin from *Garcinia mangostana*	In vitro (keratinocytes) In vivo (hairless mice)	Downregulation of UV-induced MMP-1/MMP-9 Reduction in UV-induced wrinkles	[19]
Batatasin III from *Dendrobium loddigesii*	In vitro (fibroblasts)	Increase in collagen synthesis	[20]
Ferulic acid from several plant origin	Clinical trial	Enhancement of skin elasticity by bleaching, anti-redness, smoothing, and moisturizing activities	[25]
Jasmonate derivative	In vitro (keratinocytes)	Upregulation of proteoglycans Increase in glycosaminoglycan production	[23]
Limonoids from *Carapa guianensis*	In vitro (fibroblasts)	Increase in collagen synthesis	[21]
Polysaccharides from *Panax ginseng*	In vitro (keratinocytes)	Downregulated UV-induced MMP-1 expression	[24]
Triterpenoids from *Eriobotrya japonica*	In vitro (fibroblasts)	Increase in collagen synthesis	[22]

**Table 2 molecules-27-07518-t002:** Main plant herbal products with anti-aging effects by acting on oxidative stress.

Plants	Effects	Refs
*Alchemilla mollis*	DPPH and ABTS radical scavenging activity	[36]
*Allium sativum*	DPPH radical scavenging activity Reduction in NO production Decrease in UVB-induced ROS generation	[49]
*Camelia sinensis*	DPPH, ABTS, CUPRAC, FRAP, and ORAC radical scavenging activity Decrease in UVB-induced ROS generation in fibroblasts Upregulation of SOD, CAT, and GPX in fibroblasts Increase in Nrf2 transcriptional level and nuclear translocation in fibroblasts	[38]
*Cassia fistula*	DPPH and ABTS radical scavenging activity	[43]
*Curcuma heyneana*	DPPH radical scavenging activity	[52]
*Gastrodia elata*	DPPH and ABTS radical scavenging activity Increase in SOD activity	[27]
*Hibiscus sabdariffa*	DPPH and FRAP radical scavenging activity Decrease in UVB-induced ROS generation	[53]
*Litchi chinensis*	DPPH, superoxide, and ABTS radical scavenging activity	[28]
*Magnolia officinalis*	DPPH and FRAP radical scavenging activity	[29]
*Malaxis acuminata*	DPPH and ABTS radical scavenging activity	[30]
*Manikaria zapota*	DPPH and ABTS radical scavenging activity	[14]
*Nephelium lappaceum*	DPPH, superoxide, and ABTS radical scavenging activity	[28]
*Nictanthes arbor-tristis*	DPPH radical scavenging activity Reduction in UVB-induced ROS production in human fibroblasts	[62]
*Passiflora tarminiana*	ORAC radical scavenging activity	[45]
*Penthorum chinense*	DPPH radical scavenging activity	[47]
*Phyllanthus emblica*	DPPH and ABTS radical scavenging activity	[14]
*Piper cambodianum*	Reduction in UVB-induced ROS production in human fibroblasts	[34]
*Pourthiaea villosa*	DPPH and ABTS^+^ radical scavenging activity Inhibition of H_2_O_2_-induced intracellular ROS production Increase in SOD1 and SOD2 proteins levels Inhibition of H_2_O_2_-induced premature cellular senescence	[51]
*Prosopis cineraria*	DPPH radical scavenging activity	[63]
*Salvia officinalis*	DPPH radical scavenging activity	[48]
*Spatholobus suberectus*	Reduction in UVB-induced ROS production in human fibroblasts	[32]
*Silybum marianum*	DPPH and ABTS radical scavenging activity	[14]
*Syzygium aromaticum*	Reduction in UVB-induced ROS production in human fibroblasts	[50]
*Tamarindus indica*	DPPH, superoxide, and ABTS radical scavenging activity	[28]
*Ulmus macrocarpa*	DPPH and ABTS^+^ radical scavenging activity Inhibition of H_2_O_2_-induced intracellular ROS production Increase in SOD1, SOD2, and protein levels Inhibition of H_2_O_2_-induced premature cellular senescence	[61]

ABTS = 2,2′-Azino-bis(3-ethylbenzothiazoline)-6-sulfonic acid; CAT = Catalase; DPPH = 2,2-diphenyl-1-picrylhydrazyl; H_2_O_2_ = hydrogen peroxide; NO = nitric oxide; ROS = reactive oxygen species; SOD1 = zinc-dependent superoxide dismutase; SOD2 = superoxide dismutase 2; UVB = ultraviolet B.

**Table 3 molecules-27-07518-t003:** Main plant herbal products with anti-aging effects acting on skin pigmentation.

Plants	Effects	Refs
*Cassia fistula*	In vitro inhibition of l-DOPA oxidation by tyrosinase	[43]
*Citrus junus*	In vitro inhibition of l-DOPA oxidation by tyrosinase Inhibition of melanin synthesis in melanoma cells	[39]
*Curcuma heyneana*	In vitro inhibition of l-DOPA oxidation by tyrosinase	[52]
*Eugenia dysenterica*	In vitro inhibition of l-DOPA oxidation by tyrosinase	[26]
*Hibiscus sabdariffa*	In vitro inhibition of tyrosinase activity Downregulation of MITF, tyrosinase, TRP-1, and TRP-2 gene expression Reduction in MITF, tyrosinase, TRP-1, and TRP-2 levels	[53]
*Litchi chinensis*	Reduction in melanin content in human fibroblasts Inhibition of tyrosinase activity in melanoma cells Inhibition of TRP-2 activity in melanoma cells	[28]
*Magnolia officinalis*	In vitro reduction in melanin content in CCD-966KS cells	[29]
*Malaxis acuminata*	In vitro inhibition of l-DOPA oxidation by tyrosinase	[30]
*Penthorum chinense*	Inhibition of melanin content in B16F10 cells	[47]
*Prosopis cineraria*	In vitro inhibition of l-DOPA oxidation by tyrosinase Reduction in melanin content in a clinical trial	[63]
*Tamarindus indica*	In vitro inhibition of melanin synthesis in melanoma cells Decrease in tyrosine mRNA and protein level	[28]

l-Dopa = l-3,4- dihydroxyphenylalanine; MITF = Microphthalmia-associated transcription factor; TRP-2 = Tyrosinase-related proteins-2.

## Data Availability

Not applicable.

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
