# Peer review of "Recent Advances in Herbal-Derived Products with Skin Anti-Aging Properties and Cosmetic Applications"

_molecules, 2022, doi:10.3390/molecules27217518_

Round 1

Reviewer 1 Report

This review article addresses an interesting and timely topic. I suggest its publication subject to certain revisions:

- can the authors add some experimental results adopted from already published research papers? this would make the paper more comprehensive 

- the authors need to comment on the validity of using extracts, since these are generally poorly characterised mixture of active and other compounds. 

- all biological species names should be given in Italics and following the generally accepted rules 

- some figures are of low resolution

- is it needed to add this "The illustration was drawn using BioRender software" into every figure caption?

Author Response

REVIEWER 1

Comment of reviewer: This review article addresses an interesting and timely topic. I suggest its publication subject to certain revisions:

                Answer: Thank you very much for the comments

Comment of reviewer: 1- can the authors add some experimental results adopted from already published research papers? this would make the paper more comprehensive 

                Answer: I completely understand the comments of the reviewer. However, our idea was to select, after extensive review, those manuscripts with better data related to skin aging processes. Considering that to add experimental results, in some cases, it is necessary for the authors and journal approvals, we decided to describe the general description of general results and their biological importance as potential anti-aging products, avoiding intellectual property rights problems. At the same time, all published experimental data can be accessed by visiting the original papers referred to in the bibliography references of the manuscript.

Comment of reviewer: 2- the authors need to comment on the validity of using extracts, since these are generally poorly characterised mixture of active and other compounds. 

                Answer: Thank you very much for the comments. We included in the new version of the manuscript a general comment about the biological validity of plant extracts, mainly considering their pharmacological effects and source of active ingredients for the development of newly active products.

Comment of reviewer: 3- all biological species names should be given in Italics and following the generally accepted rules 

                Answer: Thank you. We corrected those plant names that were incorrect.

Comment of reviewer: 4- some figures are of low resolution

                Answer: Thank you. Although we used the “molecules” journal rules, if our manuscript will be approved for publication, we will substitute the figures using more resolution figures.

Comment of reviewer: 5- is it needed to add this "The illustration was drawn using BioRender software" into every figure caption?

                Answer: This is a “Molecules” journal rule.

I also inform you that the English language was revised by a native person from the USA.

Reviewer 2 Report

Comments for the Authors

The review article entitled “Recent advances in herbal-derived products with skin anti-ag- 2 ing properties and cosmetic applications” is well compiled, however, it needs to be revised by addressing the following queries and suggestions.

1)      The abstract is very concise. One cannot get a complete picture of the review from the abstract, so I will suggest to modify the abstract, which containing a clear aim, results and conclusion.

2)      Please mention the databases and publishers you have used for the collection of the literature.

3)      Quality of figure 2 is very poor.

4)      The review contains articles of last how many years? Please make it clear.

Author Response

Comment of reviewer: The review article entitled “Recent advances in herbal-derived products with skin anti-aging properties and cosmetic applications” is well compiled, however, it needs to be revised by addressing the following queries and suggestions.

                Answer: Thank you very much for the comments

Comment of reviewer 1: The abstract is very concise. One cannot get a complete picture of the review from the abstract, so I will suggest to modify the abstract, which containing a clear aim, results and conclusion.

Answer: Thank you very much. We changed and improved the abstract according to these comments.

Comment of reviewer 2: Please mention the databases and publishers you have used for the collection of the literature.

Answer: Thank you very much. We included this in the new version of the manuscript

Comment of reviewer 3: Quality of figure 2 is very poor.

Answer: Thank you. Although we used the “molecules” journal rules, if our manuscript will be approved for publication, we will substitute the figures using more resolution figures.

    Comment of reviewer 4: The review contains articles of last how many years? Please make it clear.

Answer: Thank you very much. We included this in the new version of the manuscript

I also inform you that the English language was revised by a native person from the USA.

Reviewer 3 Report

Thank you very much for submitting the manuscript “Recent advances in herbal-derived products with skin anti-aging properties and cosmetic applications” in Molecules.

I appreciate your excellent review paper regarding herbal-derived products. I think this review is a big deal and it is interested at the mechanisms of the products for skin anti-ageing properties.   However, I think in vivo data is poor and this hypnosis derived review walk on your own using in vitro studies it is unclear effective evidence to human, because the main ingredients of almost herbal-derived extracts are changeable among locations, parts, seasons and exposure periods and doses of them are unknown. In addition, it may be difficult for effects of them to decide by only one reference in accordance with the systematic review.  In addition, the effective pure substances of each product are not aware how they include in the products and it is not clear correlated with the substances in session 7.  Unfortunately, this context is insufficient for the scientific journal and this one should be submitted for cosmetic applications in the non-review journal.  Thank you for your consideration.

Author Response

Comment of reviewer 1: Thank you very much for submitting the manuscript “Recent advances in herbal-derived products with skin anti-aging properties and cosmetic applications” in Molecules.

I appreciate your excellent review paper regarding herbal-derived products. I think this review is a big deal and it is interested at the mechanisms of the products for skin anti-ageing properties. 

                Answer: Thank you very much for the comments

Comment of reviewer 2: However, I think in vivo data is poor and this hypnosis derived review walk on your own using in vitro studies it is unclear effective evidence to human, because the main ingredients of almost herbal-derived extracts are changeable among locations, parts, seasons and exposure periods and doses of them are unknown.

                Answer: Thank you very much. If I understood, the reviewer commented about poor in vivo data available. Yes, this is true, but our manuscript focused on those products with in vivo studies, mainly evaluated in clinical trials. Other products without in vivo studies were included in the manuscript considering the aims of review, which is an update of natural products with skin anti-aging properties. Moreover, the reviewer is completely correct when affirms that plant-derived extracts are changeable according to environmental aspects. This is an unresolved problem for all pharmacological, chemical, and toxicological studies using plant extracts. However, these aspects do not diminish the importance of all the research carried out with natural products, which were responsible for the development of an uncountable number of new drugs and new phytotherapeutics with efficacy, safety, and quality control determined.

Comment of reviewer 3: In addition, it may be difficult for effects of them to decide by only one reference in accordance with the systematic review.  

                Answer: Yes, I agree with the reviewer, but our manuscript is to update the pharmacological data of natural products recently evaluated in several pharmacological methods related to skin aging processes and available these data as a basis for selection for plant raw material, plant extracts or isolated compounds from plants for further studies. Of course, we suggested the main isolated products and main herbal-derived products just based on a wide number of studies, including clinical trials

Comment of reviewer 4: In addition, the effective pure substances of each product are not aware how they include in the products and it is not clear correlated with the substances in session 7.  

                Answer: Our objectives are not related to this idea. I understand and appreciate the comment, but as commented, our review is important as the basis for the selection of products for further studies. It is not a study to propose these compounds or herbal-derived extracts as a final cosmetic product.

Comment of reviewer 5: Unfortunately, this context is insufficient for the scientific journal and this one should be submitted for cosmetic applications in the non-review journal.  Thank you for your consideration.

                Answer: Although I disagree, I completely respect the opinion of the reviewer and I wait for the final decision of the editors

Round 2

Reviewer 3 Report

Thank you for your replies. I agree your almost commnets. However, the authors mentioned this is true poor in vivo data, but our manuscript focused on those products with in vivo studies, mainly evaluated in clinical trials.  I ceched the revised version.  Yous manuscript is no change regaring poor in vivo data and forcused mecanisms of plan extracts mainly.  Your comments make no sense.   I hope you sart afresh for drafting a good paper.  Thank you.

Author Response

Dear reviewer

I thank you. However, I did not understand the comments probably because were unclea for me. I think the reviewer did not understand our previous answers. Yes, I confirm that our manuscript was not change regarding to in vivo data as requested. As previously commented, the in vivo (clinical trials) available data were used to select the main products with skin anti-aging properties. It is important to note that in vivo studies are generally restricted to clinical trials, and it is very uncommon the use of animals for vivo studies in the cosmetic area. In fact, in several countries the use of animals in cosmetic research is prohibited. This way, the scientific literature is poor in vivo studies, except those reported in our review. I also did not understand the request about extensive editing English language, because as commented the English Language was revised by USA native person.  I thank you very much for comments and suggestions and I will also understand if our manuscript will be not approval based on his comments. There is no problem.
